# Luminescent κ-Carrageenan-Based Electrolytes Containing Neodymium Triflate

**DOI:** 10.3390/molecules24061020

**Published:** 2019-03-14

**Authors:** S. C. Nunes, S. M. Saraiva, R. F. P. Pereira, M. M. Silva, L. D. Carlos, P. Almeida, M. C. Gonçalves, R. A. S. Ferreira, V. de Zea Bermudez

**Affiliations:** 1Chemistry Department, University of Trás-os-Montes e Alto Douro, 5001-801 Vila Real, Portugal; sofia.m.s.73@gmail.com (S.M.S.); cristina@utad.pt (M.C.G.); 2Chemistry Department, University of Beira Interior, 6201-001 Covilhã, Portugal; pjsa@ubi.pt; 3Chemistry Department, University of Minho, 4710-057 Braga, Portugal; rpereira@quimica.uminho.pt (R.F.P.P.); nini@quimica.uminho.pt (M.M.S.); 4Physics Department, University of Aveiro, 3810-193 Aveiro, Portugal; lcarlos@ua.pt (L.D.C.); rferreira@ua.pt (R.A.S.F.); 5CICS-UBI, University of Beira Interior, 6201-001 Covilhã, Portugal; 6CQ-VR, University of Trás-os-Montes e Alto Douro, 5001-801 Vila Real, Portugal

**Keywords:** κ-carrageenan, structure, biopolymer electrolyte, thermal properties, ionic conductivity, luminescence features

## Abstract

In recent years, the synthesis of polymer electrolyte systems derived from biopolymers for the development of sustainable green electrochemical devices has attracted great attention. Here electrolytes based on the red seaweeds-derived polysaccharide κ-carrageenan (κ-Cg) doped with neodymium triflate (NdTrif_3_) and glycerol (Gly) were obtained by means of a simple, clean, fast, and low-cost procedure. The aim was to produce near-infrared (NIR)-emitting materials with improved thermal and mechanical properties, and enhanced ionic conductivity. Cg has a particular interest, due to the fact that it is a renewable, cost-effective natural polymer and has the ability of gelling in the presence of certain alkali- and alkaline-earth metal cations, being good candidates as host matrices for accommodating guest cations. The as-synthesised κ-Cg-based membranes are semi-crystalline, reveal essentially a homogeneous texture, and exhibit ionic conductivity values 1–2 orders of magnitude higher than those of the κ-Cg matrix. A maximum ionic conductivity was achieved for 50 wt.% Gly/κ-Cg and 20 wt.% NdTrif_3_/κ-Cg (1.03 × 10^−4^, 3.03 × 10^−4^, and 1.69 × 10^−4^ S cm^−1^ at 30, 60, and 97 °C, respectively). The NdTrif-based κ-Cg membranes are multi-wavelength emitters from the ultraviolet (UV)/visible to the NIR regions, due to the κ-Cg intrinsic emission and to Nd^3+^, ^4^F_3/2_→^4^I_11/2-9/2_.

## 1. Introduction

The studies of solid polymer electrolytes (SPEs) based on biopolymers [1] have exponentially increased in the last few years because of the desirable properties of these macromolecules and of the growing global environmental concerns. Biopolymers offer a wide range of benefits as they are biodegradable, renewable, abundant, and non-hazardous compared to synthetic polymers. Innovative SPEs based on cellulose and its derivatives [2,3,4], deoxyribonucleic acid (DNA) [5,6], gelatin [7,8,9], chitosan [10,11], corn starch [12,13], agar [14,15], xanthan gum [16] and silk fibroin [17], were introduced. In the same context, we explored the use of the red-seaweeds-derived carrageenan (Cg) acid polysaccharides [18]. All these works revealed that these natural macromolecules have tremendous application potential [19] in various solid state electrochemical devices such as dye-sensitized solar cells [20,21], fuel cells [22,23,24], energy storage devices [3], and electrochromic devices (ECDs) [7,10,14].

Although Cgs are of the utmost interest for the energy area, the number of works reported in literature dealing with their use as SPEs is scarce [18,21,24,25,26,27,28,29,30,31]. Cgs are linear high-molecular-weight sulfated polysaccharides made up of repeating galactose units and 3,6-anhydrogalactose, both sulfated and non-sulfated [32], joined by alternating alpha 1-3 and beta 1-4 glycosidic linkages. They are water-soluble polymers extensively employed in the food industry sector, as stabilizer, thickener, emulsifier, and gelling agent. Cgs are composed of long, highly flexible molecules that curl, forming helical structures, and therefore, they have an ability to form a variety of different gels at room temperature. Depending on the method and the algae from which Cg is extracted, three main types of Cgs can be obtained: (1) Kappa (κ) (Figure 1a), that forms strong, rigid gels in the presence of potassium (K^+^) ions; (2) Iota (τ) which forms soft gels in the presence of calcium (Ca^2+^) ions; and (3) lambda (λ), which does not form gels. The primary differences that influence the gel properties of κ-Cg, τ-Cg and λ-*Cg* are the number and position of the ester sulfate groups in the repeating galactose units. The Cgs are all soluble in water, but, while λ-Cg forms viscous solutions, κ- and τ-Cgs form thermoreversible gels. In solution, the molecules of the κ and τ types undergo a thermoreversible coil-to-helix transition, where the helices self-associate via hydrogen bonds and ionic interactions, giving rise to a three-dimensional gel structure [33,34], which ultimately results in the formation of ionotropic and thermotropic gels [35,36]. The three-dimensional ordered packing of κ-Cg allows each sulfate group to be effectively surrounded by K^+^ ions, forming firm, but brittle gels, [37] while τ-Cg gelation is dependent on the presence of Ca^2+^ ions, forming soft and elastic gels [36]. κ-Cg can also form cross-linking networks with other components present in the SPEs, such as ionic liquids (ILs) [18] and lanthanide ions [30].

In 2017 we introduced κ-Cg-based biopolymer electrolytes with foreseen application in solid state electrochemical devices, as long as their operation does not require the flow of gases and does not lead to water formation, such as ECDs [18]. These attractive green flexible electrolytes were prepared in aqueous solution, by means of a simple, clean, fast and low-cost procedure, from κ-Cg, the IL 1-butyl-3-methylimidazolium chloride ([Bmim]Cl), and glycerol (Gly). The highest ionic conductivity achieved was 8.47 × 10^−4^/2.45 × 10^−3^ S cm^−1^ at 20/66 °C under anhydrous conditions, and 54.9/186 m S cm^−1^ at 30/60 °C at a relative humidity of 98%.

More recently, we investigated a 5-layer configuration ECD including (as an external layer) amorphous indium zinc oxide (a-IZO), a conducting oxide with high transmission in the visible and near-infrared (NIR) spectral regions, together with an innovative NIR-emitting electrolyte composed of κ-Cg, Gly and erbium triflate (ErTrif_3_.xH_2_O) [30]. The sample with highest ionic conductivity (1.5 × 10^−4^ S cm^−1^ at 20 °C) displayed ultraviolet (UV)/blue and NIR emissions associated with the κ-Cg based host and the Er^3+^ ions (^4^I_15/2_ → ^4^I_13/2_), respectively. The ECD tested demonstrated fast switching time (50 s), high switching efficiency (transmittance variations of 46/51% at 550/1000 nm), high optical density (0.89/0.75 at 550/1000 nm), outstanding coloration efficiency (450th cycle: −15,902/−13,400 cm^2^ C^−1^ and +3072/+2589 cm^2^ C^−1^ at 550/1000 nm for coloration and bleaching, respectively), excellent electrochemical stability, and self-healing following mechanical stress. The ECD encompassed two voltage-operated modes: semi-bright warm (+3.0 V, transmittances of 52/61% at 550/1000 nm) and dark cold (−3.0 V, transmittances of 7/11% at 550/1000 nm) [30].

In the present work, we enlarged the study of the NIR-emitting κ-Cg electrolytes to the analogue system doped with trivalent neodymium (Nd^3+^) ions, introduced as neodymium (III) triflate (NdTrif_3_) (Figure 1b). The Nd^3+^ ions are attractive in the area of SPEs owing to their high Lewis acidity and coordination number. Silva et al. prepared SPEs from poly(oxyethylene) (POE) and europium (Eu^3+^), Nd^3+^, and Er^3+^ triflates [38,39,40]. These authors reported relatively higher ionic conductivity for the system doped with NdTrif_3_ with respect to the analogue electrolytes doped with the other two lanthanide triflates [40]. As κ-Cg exhibits relatively low ionic conductivities at room temperature (in the range of 10^−7^ S cm^−1^ [18]), the addition of Gly and NdTrif_3_ allowed circumventing this problem. The surface morphology of the films was characterized by Scanning Electronic Microscopy (SEM), the structure of the films was examined by X-ray diffraction (XRD) measurements, and the thermal behavior of the films was analyzed by Differential Scanning Calorimetry (DSC). The influence of the neodymium salt concentration on the ionic conductivity values was evaluated by impedance spectroscopy. The degree of ionic association in the materials was analyzed through Fourier Transform Raman (FT-Raman) spectroscopy. The emission and excitation features were examined from the UV/visible to the NIR regions. The produced membranes were denoted as CG_x_Nd_z_, where C represents κ-Cg, G stands for Gly, and x and z indicate the concentrations of Gly and NdTrif_3_, respectively, with respect to κ-Cg.

## 2. Results and Discussion

### 2.1. Structure and Morphology

The XRD patterns of the CG_x_Nd_z_ membranes display an intense broad and non-resolved Gaussian peak located at 20.7–21.1° and weak peaks around 15.8, 18.3, 29.5 and 31.7° (Figure 2), revealing a semi-crystalline nature, with predominance of amorphous phase. In all the diffractograms of the Nd^3+^-doped κ-Cg-based membranes the sharp Bragg reflections of the pure salt are missing (Figure 2, pink line), meaning that the κ-Cg-membrane is a good matrix for the dissolution and thus encapsulation of the salt in the range of concentrations analysed.

Carrageenan gels possess thermo-reversible property, presenting somewhat crystalline regions, called junction zones, and amorphous regions. Prior to discussing the SEM data, it is useful to explain the gelation of κ-Cg in aqueous solutions during cooling by means of the zipper model [41,42,43,44]. In light of this methodology, the gelation process can be explained through two stages: (1) the polymer chains change from random coils to helices yielding clusters soluble; and (2) rigid ordered double helices are formed which then aggregate into network junctions in the presence of the so-called gelling cations, such as K^+^ and Ca^2+^, which are responsible for the occurrence of intra- and intermolecular interactions, respectively [41,42,43].

Thus, the gel-sol/sol-gel transition associated with heating/cooling corresponds to the opening/closing of zippers. In addition, the crystallinity degree of κ-Cg is correlated with the degree of packing of the helices [44]. Recently, we concluded that the CG_50_Er_0_ membrane contained micro-aggregates of variable shapes rich in intra- and intermolecular bridges (i.e., OSO_3_^− ···^ K^+^^···^ O and OSO_3_^−^
^···^ Ca^2+ ··· −^O_3_SO cross-linkages, respectively) [30].

The SEM images of the CG_50_Nd_z_ system (Figure 3) reveal that the samples display globally a homogeneous, non-rough texture. Small spherical micro-aggregates are detected in the membranes with z = 10 and 40. The analysis of the EDS mapping images of CG_50_Nd_z_ demonstrates that the different atoms are homogeneously distributed (Figure 3a and Appendix A). In the case of the analogue system incorporating ErTrif_3_, the formation of larger micro-aggregates made the detection of Ca^2+^, Er^3+^ and -OSO_3_^−^ ions possible, thus indicating the co-existence of [OSO_3_^−^
^···^ Ca^2+ ··· −^O_3_SO] and [OSO_3_^−^
^···^ Er^3+ ··· −^O_3_SO]^+^ intermolecular cross-linkages, the latter probably counter-balanced by the Trif^−^ ions [30]. The same applies here to the [OSO_3_^−^
^···^ K^+ ··· −^O_3_SO]- intramolecular bridges, which are not discerned either [30]. We recall that the commercial κ-Cg employed to synthesize the CG_50_Nd_z_ membranes contains a minor content of Ca^2+^ and K^+^ gelling ions (see Materials in Experimental section).

### 2.2. Thermal Behavior

The DSC curves of the CG_50_Nd_z_ membranes in the 25–300 °C range, reproduced in Figure 4, show an endothermic peak centered around 123–131 °C. This event is attributed to the reversible gel-sol transition temperature (T_g-s_) involving the change of the polymer chains from double helices, then to helices (soluble clusters) and ultimately to random coils. The increase of the T_g-s_ of CG_0_Nd_0_ (115.0 °C) observed upon introduction of Gly and NdTrif_3_ can be interpreted as decrease in segmental motion due to the formation of a hydrogen-bonded network between the -OH groups of Gly and the -OH and/or -C-O-C- and/or -OSO_3_^−^ polar groups of κ-Cg, and intermolecular bridges between the Nd^+^ ions and κ-Cg, which strengthen the gel. The polymer chains become harder since the rotation of the polymer segments is blocked by these cross-linkage bonds, and the flexibility of the polymer backbone is reduced. This upshift can also be explained on the basis of the zipper model. According to this model, the heat capacity of a gel depends on the number of zippers (N), on the number of parallel links (N) of a zipper, on the rotational freedom (G) of a link, and on the energy required to open a link [42]. Consequently, the T_g-s_ increase observed upon addition of NdTrif_3_ to the host κ-Cg can be associated with an increase of N and reduction of G, as a result of the aggregation of adjacent κ-Cg double helices via the Nd^3+^ ions which play the role of cross-linking agents, in a way similar to the Ca^2+^ ions. However, the T_g-s_ value of the various Nd^3+^-doped membranes remained practically unchanged (Table 1 and Figure 4). The reason for this effect could be the occurrence of a high number of intermolecular cross-linkages all over the materials leading to the formation of many separate junction zones each involving a few double helices. This process would yield small micro-aggregates, as suggested by the SEM data.

All the DSC curves also present an exothermic peak above 190 °C (Appendix A and Table 1) associated with the thermal decomposition (T_d_) of the κ-Cg [18]. The analysis of Figure 4b leads us to conclude that doping the CG_50_Nd_z_ matrix with low contents of NdTrif_3_ increased the stability of the biopolymer matrix. However, the increase of NdTrif_3_ content for z ≥ 20 destabilized the system, shifting the onset of degradation to lower temperatures.

### 2.3. Ionic Conductivity Study

The degree of salt dissociation, salt concentration, ion mobility, the dielectric constant of the host polymer, and the segmental mobility of the polymer chains [45,46] influence the ion transport properties of SPEs. In the present work, Gly was added as plasticizer to increase the amorphous phase content; dissociate ionic aggregates, and lower the glass transition temperature [45] of the CG_50_Ndz membranes.

Figure 5 shows the Arrhenius conductivity plot of the CG_50_Nd_z_ membranes in the 20–110 °C temperature range at variable concentration of NdTrif_3_. Below the T_g-s_ value all the doped samples demonstrate a non-linear variation of the ionic conductivity with temperature, a behavior typically found in disordered electrolytes. This process is favoured in the presence of the plasticizer which primarily increases the fraction of free volume by better separation of the polymer chains and ultimately influences the movement of charge carriers. The examination of this plot reveals that the membrane with the highest conductivity in the temperature range studied is CG_50_Nd_20_. This sample exhibits 1.03 × 10^−4^, 3.03 × 10^−4^, and 1.69 × 10^−4^ S cm^−1^ at 30, 60, and 97 °C, respectively (Figure 5, green symbols). With the increase of salt content (z = 30), a marked reduction in conductivity is observed (Figure 5, blue symbols). At this salt composition the salt begins to be poorly dissociated and ion aggregates probably form. As a consequence, the number and mobility of charged species present in the electrolyte system is reduced at high salt content and hence the conductivity decreases. However, at z = 40 the ionic conductivity suffers a slight increase. We will return to the analysis of the concentration dependence of conductivity in the section devoted to the FT-Raman analysis. Above the T_g-s_ value the ionic conductivity of most samples suffered a marked decrease. This effect can be ascribed to the dramatic loss of the mechanical properties of the membranes at temperatures higher than T_g-s_.

It is worth comparing the present results with those reported elsewhere for CG_50_Er_z_ electrolytes [30] and POE/Nd^3+^ [40]. The conductivity values of the present system are similar to those reported for SPEs based on κ-Cg and ErTrif_3_ (1.5 × 10^−4^ and 3.6 × 10^−4^ S cm^−1^ at 20 and 60 °C, respectively) [30], or POE and NdTrif_3_ (3.16 × 10^−4^ S cm^−1^ at 30 °C) [40].

### 2.4. Ionic Association Study

To evaluate the ionic association in the Nd^3+^-doped κ-Cg-based membranes, the symmetric stretching vibration mode of the SO_3_ group (ν_s_SO_3_) was studied through the analysis of the FT-Raman spectra. Several species are known to exist in SPEs: (a) “free” or weakly bonded ions with considerable mobility; (b) cations strongly bonded to the host polymer and thus with low mobility; (c) ionic aggregates, such as contact ion pairs and higher ionic multiplets, with low-moderate mobility.

The FT-Raman spectra of the CG_50_Nd_z_ membranes in the ν_s_SO_3_ region is represented in Figure 6a. Because the ν_s_SO_3_ band is superimposed with that due to the stretching vibration mode of the S=O group of the sulfate ester (-OSO_3_H unit) of κ-Cg, at 1063 cm^−1^ [47], it was necessary to first subtract the FT-Raman spectrum of the matrix from those of the Nd^3+^-doped κ-Cg-based membranes. Figure 6b shows the results of the curve-fitting performed in the subtracted ν_s_SO_3_ FT-Raman band.

In short, the ν_s_SO_3_ FT-Raman band of the CG_50_Nd_z_ membranes with z ≤ 20% was resolved into three components: a sharp band at 1031 cm^−1^ assigned to ‘‘free’’ ions and two shoulders around 1037 and 1026 cm^−1^ (Figure 6b), attributed to weakly coordinated triflate ions located in two different anionic environments [48]. In the CG_50_Nd_30_ membrane, a new component at 1042 cm^−1^ emerges (Figure 5b). This component is tentatively attributed to the formation of a crystalline compound of unknown composition and stoichiometry [48]. The presence of a crystalline complex in CG_50_Nd_30_ could be the reason for the drop of ionic conductivity in this membrane. In the case of the sample with z = 40, this component is not detected, but the concentration of species which produce the 1026 cm^−1^ band is much higher. This might explain the higher ionic conductivity of this electrolyte with respect to CG_50_Nd_30_. The spectroscopic analysis carried out provides evidence that, as expected, some of the charge carriers of membrane with highest conductivity (CG_50_Nd_20_) are very likely “free” Trif^−^ ions or weakly coordinated species.

### 2.5. UV/Visible and NIR Analysis

The κ-Cg-based membranes are multi-wavelength emitters from the UV/visible to the NIR region, as demonstrated in Figure 7a,b for two selected samples. The emission results from the overlap of a series of NIR straight lines ascribed to the Nd^3+^, ^4^F_3/2_→^4^I_11/2-9/2_ transitions with a broad band in the UV/visible spectral region attributed to the κ-Cg intrinsic emission [49], whose emission peak position deviates to the red as the excitation wavelength increases. The emission energy dependence on the excitation wavelength indicates a large distribution of emitting centres, in good agreement with the amorphous local structure of k-Cg evidenced by XRD (Figure 2). The excitation spectra were monitored around the Nd^3+^ most intense transition revealing a broad band peaking at 350 nm and a series of low-relative intensity intra-4*f*
^3^ lines arising from transitions between the ^4^I_9/2_ excited state and the ^4^D_5/2-1/2_, ^2^I_11/2_, ^2^L_15/2_, ^4^G_11/2-5/2_, ^2^G_9/2-7/2_, ^2^K_13/2_, ^4^S_3/2_ and ^4^F_7/2_ levels. We note that an analogous broad band around 350 nm also dominates the excitation spectrum monitored within the κ-Cg intrinsic emission, despite the presence of a low-relative intensity one in the low-wavelength region. The presence of the more intense excitation band found in the excitation spectra monitored within the host emission than in that monitored within the Nd^3+^ levels points out the occurrence of effective κ Cg-to-Nd^3+^ energy transfer. The fact that the κ-Cg-related band is more intense than those of the intra-4*f*
^3^ transitions in the excitation spectra monitored around the Nd^3+^ emission, readily indicates that the ions’ excited states are mainly populated though the ligands sensitisation rather than by direct intra-4f^3^ excitation. Moreover, the observation in the excitation spectra of Figure 7c,d of self-absorptions in the κ-Cg emission indicates the presence of κ-Cg-to-Nd^3+^ radiative energy transfer, i.e., the κ-Cg emission resonant with the Nd^3+^ intra-4f lines is absorbed by the metal ions and, subsequently, converted into f–f emission. Such radiative energy transfer was previously observed in Nd^+3^ and received the designation of “inner filter” effect [50].

## 3. Experimental Section

### 3.1. Materials

κ-Cg (Carrageenan CG-130, Genugel, CP Kelco, 3 and 1.3 wt.% of K^+^ and Ca^2+^, respectively) [51], neodymium (III) triflate (NdTrif, Aldrich, Steinheim, Germany, 98%), and glycerol (Gly, Sigma-Aldrich, Steinheim, Germany, 99%) were used as received. High purity deionized water (H_2_O) (type II pure water, using Elix Reference Water Purification System 10 from Millipore, Rephile Bioscience, Lda., Boston, MA, USA) was used in all experiments.

### 3.2. Preparation of the κ-Cg -Based Electrolytes

A mass of approximately 0.30 g of κ-Cg was dispersed in 15 mL of distilled water and heated under magnetic stirring at 60–70 °C during 1 h for complete dissolution. A volume of 122 μL of Gly (corresponding to 50% wt. Gly/κ-Cg) and different amounts of NdTrif_3_ were then added to this solution under stirring (Table 2). When the solutions became homogeneous, they were transferred to Petri plates and cooled down to room temperature. All the membranes prepared were stored in an oven at 50 °C over 4 days. Under these drying conditions, the colour of κ-Cg does not tend to turn brown and not observed cleavage of the glycosidic linkage [52]. The main problem associated with drying κ-Cg is the formation of a gel between this polysaccharide and water which inhibits the diffusion of water to the surface [52]. The as-produced membrane films were denoted as CG_x_Nd_z_, where C represents κ-Cg, G stands for Gly, and x and z indicate the concentrations of Gly and NdTrif_3_, respectively, with respect to κ-Cg.

### 3.3. Characterization Techniques

The X-ray diffraction (XRD) measurements were recorded at room temperature with a Rigaku Dmax III/C X-ray diffractometer, power 40 kV/30 mA and using monochromated CuK_α_ radiation (λ = 1.5418 Å) over the 2θ range of 10 to 70.0° at 1.2° min^−1^.

DSC curves of the samples were recorded using a Netzsch instrument (model STA 449 F3 Jupiter, Nurnberg, Germany). The samples were transferred to aluminum crucibles, covered with pin holed seals, and then heated from 30 to 260 °C at 10 °C min^−1^. Dry nitrogen was used as purge and protective gases (50 mL min^−1^).

Scanning electron microscopy (SEM) images were obtained at 20 kV on a Hitachi S-3400N type II microscope equipped with a Bruker x-flash 5010 at high vacuum. The sample was coated with gold. Elemental mapping of the samples was performed by Energy Dispersive X-ray (EDX) analysis. The acquisition time for a satisfactory resolution and noise performance was 30 s.

The Fourier Transform Raman (FT-Raman) spectra were recorded at room temperature with an FT Raman Bruker RFS 100/S spectrometer equipped (Hamburg, Germany) with a Nd-YAG with wavelength 1064 nm (350 mW). The spectra were collected over the 4000–100 cm^−1^ range by averaging 1500 scans at a resolution of 4 cm^−1^.

To evaluate complex FT-Raman band envelopes and to identify underlying spectral components, the iterative least-squares curve-fitting procedure in the PeakFit software (version 4) [53] was used extensively. The best fit of the experimental data was obtained by varying the frequency, bandwidth, and intensity of the bands. As the morphology of the materials was under investigation, Gaussian band shapes were employed. A linear baseline correction with a tolerance of 0.2% was used. The standard errors of the curve-fitting procedure were less than 0.0002.

Bulk ionic conductivities (σ_i_) of the membranes were obtained during heating cycles from room temperature to 60 °C, over the frequency range of 65 kHz to 0.5 Hz, by means of an Autolab PGSTAT-12 (Eco Chemie, Utrecht, The Netherlands), and using cell GE/electrolyte membrane/GE (where GE stands for 10 mm diameter ion-blocking gold electrodes, Goodfellow, > 99.95%). Prior to characterization, the κ-Cg based electrolytes were vacuum-dried at 50 °C for about 48 h and stored in an argon-filled glovebox. The electrode–membrane–electrode assembly was secured in a suitable constant-volume support, which was installed in a Buchi TO 51 tube oven. A calibrated type-K thermocouple placed close to the membrane disc was used to measure the sample temperature with a precision of about 0.2 °C. The CG_x_Nd_z_ electrolytes demonstrated almost ideal semiconductor behavior up to 60 °C and bulk conductivities were extracted in the conventional manner from impedance data by using an equivalent circuit composed of R_b_ in parallel with G_c_, where R_b_ is the bulk electrical resistance of the electrolyte and G_c_ is its geometric capacity. The circuit element corresponding to the blocking electrode interface was simulated by a series C_dl_ elements, where C_dl_ is the double layer capacity. The σ_i_ was calculated using the expression,
(1)$$\sigma i=d/\left(R_{b}\;A\right)$$
where *R_b_*, *d* and *A* are the bulk resistance, the thickness and the area of the electrolyte sample, respectively.

The emission and excitation spectra were recorded using a Fluorolog3^®^ Horiba Scientific (Model FL3-2T, Montpellier, France) spectroscope, with a modular double grating excitation spectrometer (fitted with a 1200 grooves/mm grating blazed at 330 nm) and a TRIAX 320 single emission monochromator (fitted with a 1200 grooves/mm grating blazed at 500 nm, reciprocal linear density of 2.6 nm mm^−1^), coupled to R928 (UV/visible measurements) or H10330A (NIR measurements) Hamamatsu photomultiplier, using the front face acquisition mode. The excitation source was a 450 W Xe arc lamp. The emission spectra were corrected for detection and optical spectral response of the spectrofluorimeter and the excitation spectra were corrected for the spectral distribution of the lamp intensity using a photodiode reference detector. Moreover, a band path filter was used to avoid stray radiation arising from the monitoring wavelength in the excitation spectra.

## 4. Conclusions

SPEs consisting of a κ-Cg, NdTrif_3_, and Gly were prepared via solvent-casting. In the present work, we have reported structural, morphological and optical properties, thermal behavior, ionic conductivity, and the degree of ionic associations of membranes, as a function the NdTrif_3_ content. The analysed membranes show a semi-crystalline nature, with predominance of amorphous phase and a homogeneous, non-rough texture with small spherical micro-aggregates. A maximum ionic conductivity was achieved in an optimized sample with 50 wt.% Gly/κ-Cg and 20 wt.% NdTrif_3_/κ-Cg, (1.03 × 10^−4^, 3.03 × 10^−4^, and 1.69 × 10^−4^ S cm^−1^ at 30, 60, and 97 °C, respectively). FT-Raman spectroscopy suggests that the charge carriers of membrane with highest conductivity are ‘‘free’’ Trif^−^ ions or weakly coordinated species. The NdTrif_3_-based κ-Cg membranes present UV/visible and NIR emission associated with the κ-Cg based host and the Nd^3+^, ^4^F_3/2_→^4^I_11/2-9/2_, respectively. These membranes also display radiative energy transfer, named as ‘‘inner filter’’ effect. The encouraging results reported in this work suggest that similar electrolytes incorporating highly efficient Nd^3+^ β-diketonate complexes instead of the salt employed here, may find application in ECDs featuring attractive attributes, such as continuous NIR emission and UV harvesting ability. These characteristics are of interest for the next generation of nearly-zero smart windows of the future sustainable buildings. These devices will help curbing the energy consumption in the building, will avoid the need for anti-UV coatings, while contributing to the increase of the occupant’s well-being. Considering that the transmission of this type of polysaccharide-based electrolytes is not very high in the bleached state [30], we may further speculate that this sort of electrolytes will be suitable for anti-glare purposes.

## Figures and Tables

**Figure 1 molecules-24-01020-f001:**
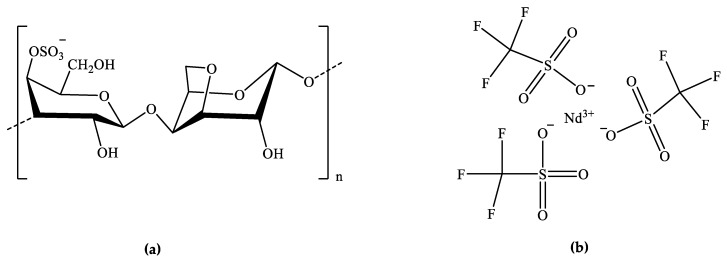
Representation of the chemical structures of κ-carrageenan (κ-Cg) (**a**) and NdTrif_3_ (**b**).

**Figure 2 molecules-24-01020-f002:**
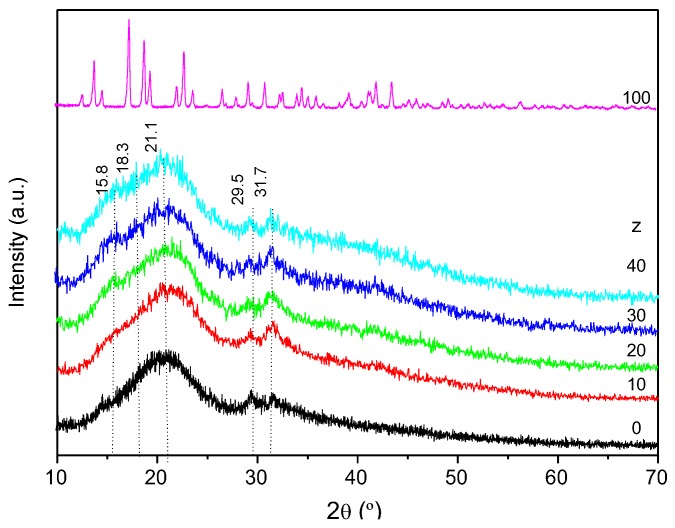
XRD curves of the non-doped (black line) and doped CG_50_Nd_z_ membranes, and of NdTrif_3_ (pink line).

**Figure 3 molecules-24-01020-f003:**
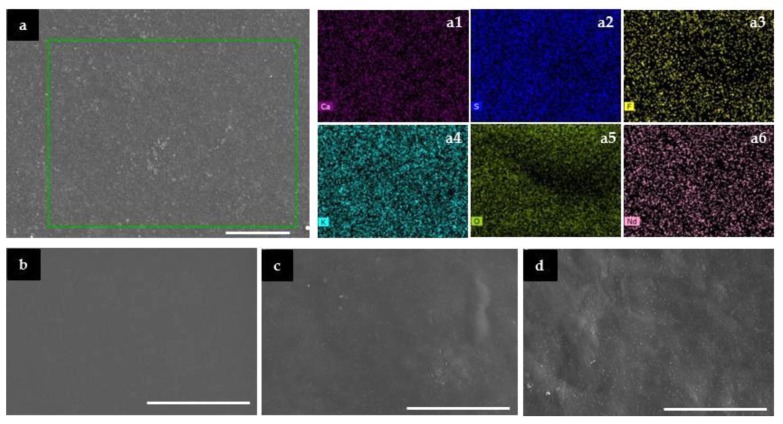
SEM images of the CG_50_Ndz membranes with z = 10% (**a**), 20% (**b**), 30% (**c**) and 40% (**d**). Scale bars = 50 μm. **a1**–**a6**: EDS mapping of the inset region of **a**.

**Figure 4 molecules-24-01020-f004:**
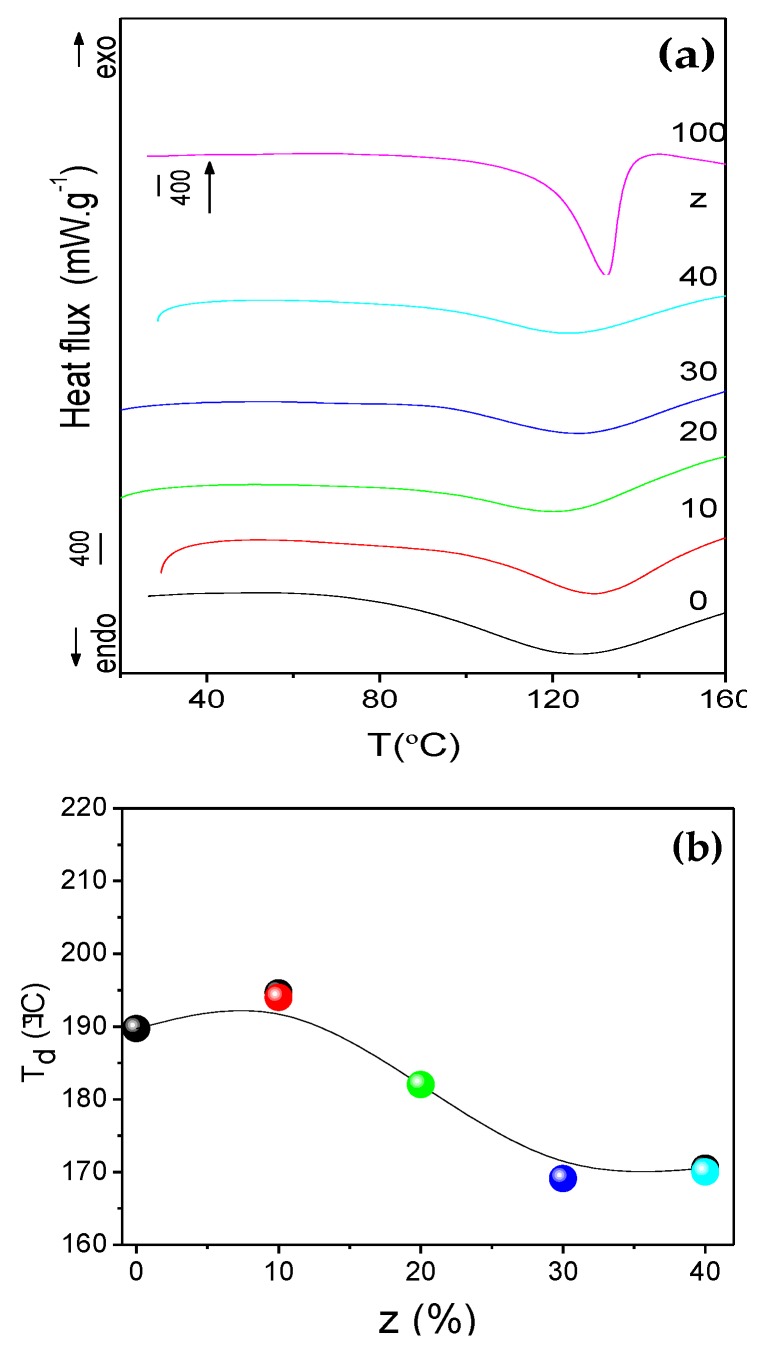
Differential Scanning Calorimetry (DSC) curves (**a**) and variation of the decomposition temperature (T_d_) with salt content z (**b**) of the non-doped and doped CG_50_Nd_z_ membranes and of NdTrif_3_ (green line). The line drawn in (**b**) is a guide for the eyes.

**Figure 5 molecules-24-01020-f005:**
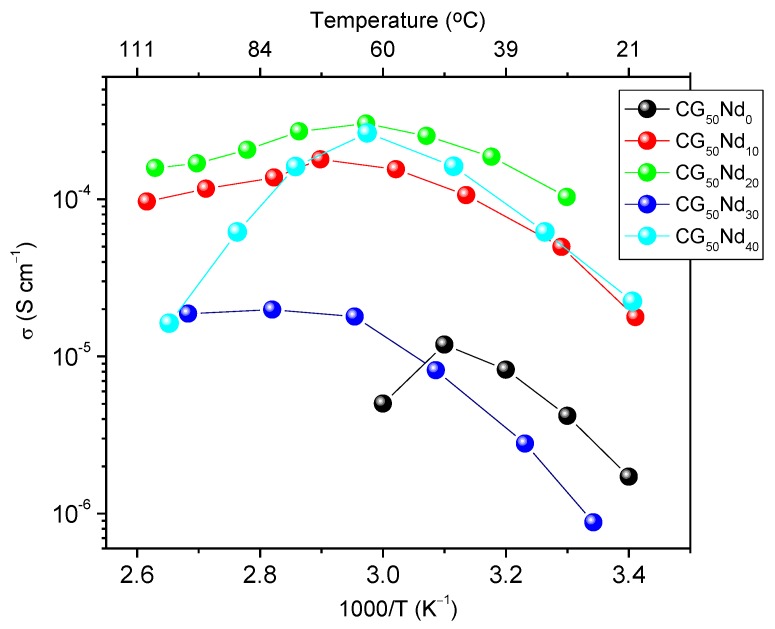
Arrhenius conductivity plot of the CG_50_Nd_z_ membranes.

**Figure 6 molecules-24-01020-f006:**
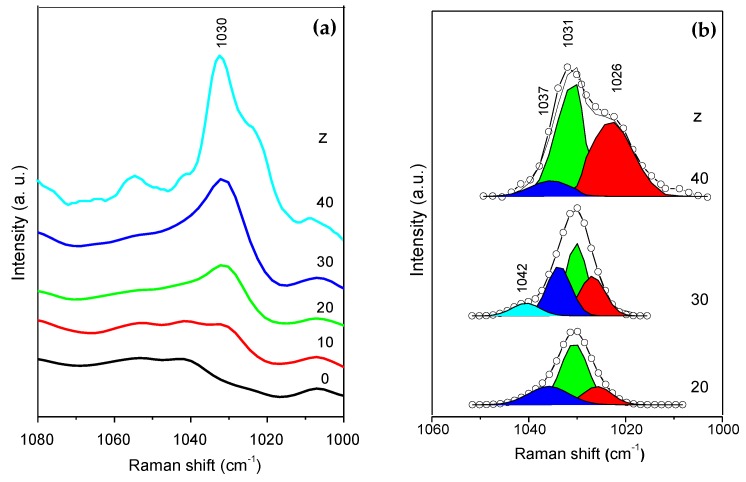
FT-Raman spectra (**a**) and curve-fitting results performed with the subtrated FT-Raman spectra (**b**) of the CG_50_Nd_z_ membranes in the ν_s_SO_3_ region.

**Figure 7 molecules-24-01020-f007:**
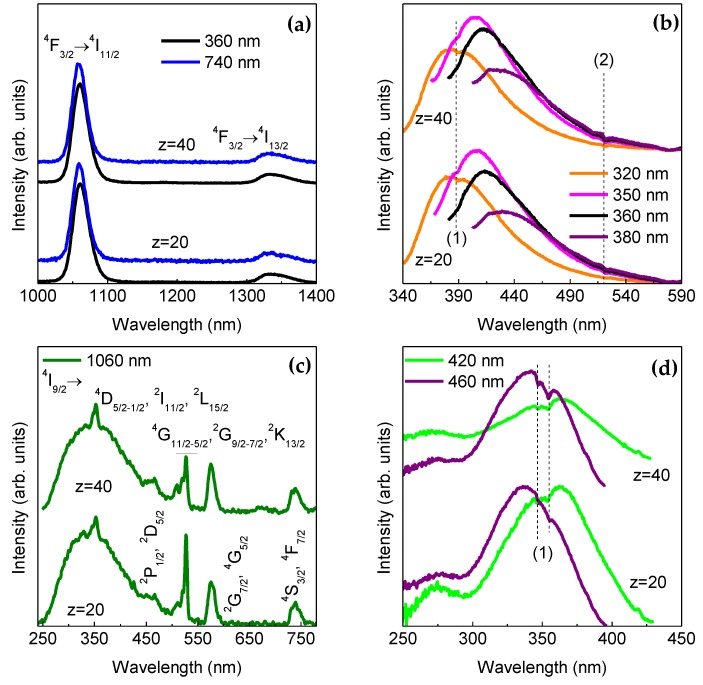
Room temperature (**a**) NIR and (b) UV/visible emission spectra and (**c**,**d**) excitation spectra of CG_50_Nd_z_, z = 20 and 40. The (**a**,**b**) excitation and (**c**,**d**) monitoring wavelengths are indicated in the figures. (1) and (2) denote the self-absorptions. ^4^I_9/2_→^4^D_5/2-1/2_, ^2^I_11/2_, ^2^L_15/2_
^4^D_5/2-1/2_, ^2^I_11/2_, ^2^L_15/2_, and ^4^I_9/2_→^4^G_11/2-5/2_, ^2^G_9/2,7/2_, ^2^K_13/2_, respectively.

**Table 1 molecules-24-01020-t001:** Gel-sol transition temperature (T_g-s_) and decomposition temperature (T_d_) of the CG_50_Nd_z_ membranes.

Sample	z (%)	T_g-s_ (°C)	T_d_ (°C)
κ-Cg	-	139 [18]	188 [18]
CG_0_Nd_0_	0	115 [18]	197 [18]
CG_50_Nd_0_	0	124	190
CG_50_Nd_10_	10	131	195
CG_50_Nd_20_	20	124	182
CG_50_Nd_30_	30	129	169
CG_50_Nd_40_	40	124	170

**Table 2 molecules-24-01020-t002:** Relevant details of the CG_x_Nd_z_ membranes.

CG_x_Nd_z_					
x	z	m (κ-Cg) (g)	V (H_2_O) (mL)	% Nd/κ-Cg	m (NdTrif) (g)
50	0	0.3017	15	-	-
	10	0.3013		10	0.0316
	20	0.3020		20	0.0615
	30	0.3006		30	0.0919
	40	0.3020		40	0.1211

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
