# Peer review of "Luminescent κ-Carrageenan-Based Electrolytes Containing Neodymium Triflate"

_molecules, 2019, doi:10.3390/molecules24061020_

Round 1
Reviewer 1 Report
In the optical properties section (figure 7a), the emission transition at 1330 nm has been denoted as transtion from 4F3/2 to 4I9/2. Is it correct?
Figure 7c, the sharp line at 530 nm may have contribution from second order stray signals as the monitoring wavelength is 1060 nm. this may be a reason why the 530 nm peak has different intensities in two samples. It is better to use optical filter to eliminate such stray signals.
Figure 7b: why the emission is different for different excitations? are there multiple emitting centers?
In the PLE spectra, (figure 7c,d), the PLE for polymer host exhibits two broad bands peaking at about 350 nm and 275 nm. However, for Nd3+, the broad PLE band in uv range exhibit only one The language part need improvement.
Author Response
Reviewer 1
Comment 1
In the optical properties section (figure 7a), the emission transition at 1330 nm has been denoted as transtion from 4F3/2 to 4I9/2. Is it correct?
Answer
The reviewer is correct. The transition is 4F3/2 to 4I13/2.
Changes included
The figure was corrected.
Comment 2
Figure 7c, the sharp line at 530 nm may have contribution from second order stray signals as the monitoring wavelength is 1060 nm. this may be a reason why the 530 nm peak has different intensities in two samples. It is better to use optical filter to eliminate such stray signals.
Answer
A band path filter was used to avoid such stray radiation and this information was added to the experimental section.
Changes included
We introduced new information to the experimental section.
“Moreover, a band path filter was used to avoid stray radiation arising from the monitoring wavelength in the excitation spectra.”
Comment 3
Figure 7b: why the emission is different for different excitations? are there multiple emitting centers?
Answer
The reviewer comment is pertinent. The emission energy dependence on the excitation wavelength indicates a distribution of emitting centers in close local environments, in good agreement with the amorphous local structure of the k-Cg (evidenced by XRD, Figure 2).
Changes included
Figure 7b discussion was improved in the revised version.
“The emission energy dependence on the excitation wavelength indicates that a large distribution of emitting centers, in good agreement with the amorphous local structure of the κ-Cg evidenced by XRD (Figure 2). “
Comment 4
In the PLE spectra, (figure 7c,d), the PLE for polymer host exhibits two broad bands peaking at about 350 nm and 275 nm. However, for Nd3+, the broad PLE band in uv range exhibit only one The language part need improvement.
Answer
We follow the reviewer comment and the language was improved.
Changes included
The language has been improved
“We note that an analogous broad band around 350 nm also dominates the excitation spectrum monitored within the carrageenan intrinsic emission, despite the presence of a low-relative intensity one in the low-wavelength region. The presence of the more intense excitation band found in the excitation spectra monitored within the host emission in that monitored within the Nd3+ levels, points out that the presence of effective carrageenan-to-Nd3+ energy transfer. The fact that the carrageenan-related band is more intense than the intensity of the intra-4f 3 transitions in the excitation spectra monitored around the Nd3+ emission, readily indicates that the ions’ excited states are mainly populated though the ligands sensitisation rather than by direct intra-4f3 excitation. Moreover, the observation in the excitation spectra of Figure 7(c,d) of self-absorptions in the Cg emission indicates the presence of Cg-to-Nd3+ radiative energy transfer, i.e., the Cg emission resonant with the Nd3+ intra-4f lines is absorbed by the metal ions and, subsequently, converted into f–f emission. Such radiative energy transfer has been previously observed in Nd+3 and received the designation of ‘‘inner filter’’ effect [50]. “
Reviewer 2 Report
In this work, red seaweeds-derived polysaccharide κ-Carrageenan (κ-Cg) were doped into neodymium triflate and glycerol to prepare near-infrared (NIR)-emitting materials. The prepared materials were characterized using X-ray diffraction, differential scanning calorimetry, scanning electron microscopy, fourier transform raman spectroscopy, impedance spectroscopy, and photoluminescence spectroscopy. I found this work has interesting results. However, several major issues have to be taken into account and address for this manuscript. Few main concerns are considered as follows:1- Abstract is too long and it writes like an introduction. However, similar sentences in Abstract repeat in the introduction. It is highly recommended the extra explanations can be moved to the introduction.
2- In abstract authors used: “Fourier Transform Raman …” with capital letter.! If authors want to use capital letter their abbreviations also need to use as well.
3- Several typos are in the manuscript. For example, in line 260 using H2O should be correct as H2O by using subscript for the number. Using correct degree Celsius (º) should be replaced to wrong used one (º) in the manuscript.
4- Authors used Experimental Section after result and discussion. This is an unusual format. Because after explaining the materials and Characterization techniques, result and discussion can be easily understood.
5- As shown in Figure 5, CG50Nd20 shows the maximum Arrhenius conductivity in comparison with others. But, for CG50Nd30 and CG50Nd40 an unusual behavior can be observed. Authors did not explain the concrete reason for this behavior. However, the authors expressed about poor dissociation and ion aggregation. It seems these reasons cannot be explained here due to increasing the higher conductivity of CG50Nd40 than CG50Nd30. More explanation can illuminate this vague.
Author Response
Comment 1
Abstract is too long and it writes like an introduction. However, similar sentences in Abstract repeat in the introduction. It is highly recommended the extra explanations can be moved to the introduction.
Answer
The abstract was written as described in the document “instructions for the authors”. However, we decided improved the abstract.Changes included
In last year’s, Recently the community of solid polymer electrolytes has turned its attention to the synthesis of polymer electrolyte systems derived from biopolymers targeting for the development of sustainable green electrochemical devices has attracted great attention. The advantage of using this type of materials is that they are non-toxic, biodegradable, renewable, abundant, and non-hazardous compared to synthetic polymers. In the present work, Here electrolytes based on the red seaweeds-derived polysaccharide κ-carrageenan (κ-Cg) doped with neodymium triflate (NdTrif3) and glycerol (Gly) were obtained by means of a simple, clean, fast and low-cost procedure. The aim was to produce near-infrared (NIR)-emitting materials with improved thermal and mechanical properties, and enhanced ionic conductivity. Cg have a particular interest, due to the fact is a renewable, cost-effective natural polymer and have the ability of gelling in the presence of certain alkali- and alkaline-earth metal cations, being good candidates as host matrices for accommodating guest cations. The materials were characterized using X-ray diffraction, differential scanning calorimetry, scanning electron microscopy, Fourier Transform Raman spectroscopy, impedance spectroscopy and photoluminescence spectroscopy. The as-synthesised κ-Cg-based membranes are semi-crystalline, reveal essentially a homogeneous texture and exhibit ionic conductivity values 1-2 orders of magnitude higher than those of the κ-Cg matrix. A maximum ionic conductivity was achieved for 50 wt.% Gly/κ-Cg and 20 wt.% NdTrif3/κ-Cg (1.03 x 10-4, 3.03 x 10-4, and 1.69 x 10-4 S cm-1 at 30, 60, and 97 ºC, respectively). The NdTrif-doped κ-Cg membranes are multi-wavelength emitters from the ultraviolet (UV)/visible to the NIR regions, attributed due to the κ-Cg intrinsic emission and to Nd3+, 4F3/2®4I11/2-9/2.
(highlighted red - to remove; highlighted yellow - to change)
Comment 2
In abstract authors used: “Fourier Transform Raman …” with capital letter.! If authors want to use capital letter their abbreviations also need to use as well.
Answer
Acknowledged.
Changes included
None.
Comment 3
Several typos are in the manuscript. For example, in line 260 using H2O should be correct as H2O by using subscript for the number. Using correct degree Celsius (º) should be replaced to wrong used one (º) in the manuscript.
Answer
We thank the referee for noticing the mistake.
Changes included
The correction has been introduced:
Comment 4
Authors used Experimental Section after result and discussion. This is an unusual format. Because after explaining the materials and Characterization techniques, result and discussion can be easily understood.
Answer
We present the Experimental Section after the Result and discussion, because we followed the “instructions for the authors”.
Changes included
None.
Comment 5
As shown in Figure 5, CG50Nd20 shows the maximum Arrhenius conductivity in comparison with others. But, for CG50Nd30 and CG50Nd40 an unusual behavior can be observed. Authors did not explain the concrete reason for this behavior. However, the authors expressed about poor dissociation and ion aggregation. It seems these reasons cannot be explained here due to increasing the higher conductivity of CG50Nd40 than CG50Nd30. More explanation can illuminate this vague.
Answer
We thank the reviewer for this remark.
Changes included
Figure 5 shows the Arrhenius conductivity plot of the CG50Ndz membranes in the 20-110 ºC temperature range at variable concentration of NdTrif3. Below the Tg-s value all the doped samples demonstrate a non-linear variation of the ionic conductivity with temperature, a behavior typically found in disordered electrolytes. This process is favored in the presence of the plasticizer which primarily increases the fraction of free volume by better separation of the polymer chains and ultimately influences the movement of charge carriers. The examination of this plot reveals that the membrane with the highest conductivity in the temperature range studied is CG50Nd20. This sample exhibits 1.03×10−4, 3.03×10−4, and 1.69×10−4 S cm−1 at 30, 60, and 97 ºC, respectively (Figure 5, green symbols). With the increase of salt content (z = 30), a marked reduction in conductivity is observed (Figure 5, blue symbols). At this salt composition the salt begins to be poorly dissociated and ion aggregates probably form. As a consequence the number and mobility of charged species present in the electrolyte system is reduced at high salt content and hence the conductivity decreases. However, at z = 40 the ionic conductivity suffers a slight increase. We will return to the analysis of the concentration dependence of conductivity in the section devoted to the FT-Raman analysis. Above the Tg-s value the ionic conductivity of most samples suffered a marked decrease. This effect can be ascribed to the dramatic loss of the mechanical properties of the membranes at temperatures higher than Tg-s.
In the section 2.4 the following new text has been added:
“The presence of a crystalline complex in CG50Nd30 could be the reason for the drop of ionic conductivity in this membrane. In the case of the sample with z = 40, this component is not detected, but the concentration of species which produce the 1026 cm-1 band is much higher. This might explain the higher ionic conductivity of this electrolyte with respect to CG50Nd30. The spectroscopic analysis carried…”
Reviewer 3 Report
<Journal Name> Molecules
Manuscript ID: molecules-454796
Type of manuscript: Article
Title: Luminescent κ-Carrageenan-based electrolytes containing neodymium triflate
Authors: S. C. Nunes *, S. M. Saraiva, R. F. P. Pereira, M. M. Silva, L. Carlos, P. Almeida, M.C. Gonçalves, R. A. S. Ferreira, V. de Zea Bermudez *
Reviewer's comment:
Dear Editor:
The manuscript focused on the on the polysaccharide κ-Carrageenan (κ-Cg) doped with neodymium triflate, which is very interesting and useful. It is recommended to accept after major revision. However, some parts need to revise, which are listed below as follows. The main points need to revise before publication.
[1] The new relate references are needed to add in the revised manuscript.
[2] What are the important applications in this study? Please add in the revised manuscript.
[3] The authors investigate many parameters in this study. What is optimal condition in this work? Please explain and add it in the revised manuscript.
[4] In Fig. 5, the Tg-s value all the doped samples demonstrate a non-linear variation of the ionic conductivity with temperature, a behavior typically found in disordered electrolytes. Please explain the reason in details.
[5] What is “inner filter’’ effect? Please explain it in details.
Sincerely yours.
Author Response
Comment 1
The new relate references are needed to add in the revised manuscript.
Answer
We accept the comment of the reviewer.
Changes included
New references were introduced.
[1] Sudhakar, Y.N.; Selvakumar, M.; Bhat, K., Biopolymer Electrolytes -Fundamentals and Applications in Energy Storage. 1st Edition ed.; 2018.
[16] Tavares, F.C.; Dörr, D.S.; Pawlicka, A.; Oropesa Avellaneda, C., Microbial origin xanthan gum-based solid polymer electrolytes. J Appl Polym Sci 2018, 135, 46229.
[19] Silva, M.M.; Bermudez, V.d.Z.; Pawlicka, A., Application of Polymer Electrolytes for Electrochemical Devices. In Polymer Electrolytes : Characterization and Applications, Winie, T.; Arof, A. K.; Thomas, S., Eds. Wiley-VCH: 2019; Vol. Volume 2.
[23] Boopathi, G.; Pugalendhi, S.; Selvasekarapandian, S.; Premalatha, M.; Monisha, S.; Aristatil, G., Development of proton conducting biopolymer membrane based on agar–agar for fuel cell. Ionics 2017, 23, 2781-2790.
[31] Liew, J.W.Y.; Loh, K.S.; Ahmad, A.; Lim, K.L.; Wan Daud, W.R., Effect of Modified Natural Filler O-Methylene Phosphonic κ-Carrageenan on Chitosan-Based Polymer Electrolytes. Energies 2018, 11, 1910.
Comment 2
What are the important applications in this study? Please add in the revised manuscript.
Answer
The foreseen application of the electrolytes developed in this work was indicated in the last sentence of the Conclusions section. Maybe we did not make our statement clear enough.
Changes included
The last sentence of the Conclusions section has been rewritten:
“The encouraging results reported in this work suggest that similar electrolytes incorporating highly efficient Nd3+ b-diketonate complexes instead of the salt employed here, may find application in ECDs featuring attractive attributes, such as continuous NIR emission and UV harvesting ability. These characteristics are of interest for the next generation of nearly-zero smart windows of the future sustainable buildings. These devices will help curbing the energy consumption in the building, will avoid the need for anti-UV coatings, while contributing to the increase of the occupant’s well-being. Considering that the transmission of this type of polysaccharide-based electrolytes is not very high in the bleached state [30], we may further speculate that this sort of electrolytes will be suitable for anti-glare purposes.”
Comment 3
The authors investigate many parameters in this study. What is optimal condition in this work? Please explain and add it in the revised manuscript.
Answer
When we are dealing with a polymer electrolyte the most critical parameters to consider are the composition (here the contents of Gly and lanthanide salt), the amorphous character, the mechanical properties and especially the ionic conductivity.
Changes included
To make clear to the reader that we were looking for the composition that would yield the highest ionic conductivity, we have rephrased the sentence
“A maximum ionic conductivity was achieved with50 wt.% Gly/κ-Cg and 20 wt.% NdTrif3/κ-Cg, (1.03 x 10−4, 3.03 x 10−4, and 1.69 x 10−4 S cm−1 at 30, 60, and 97 ºC, respectively).”
was replaced by
“A maximum ionic conductivity was achieved in an optimized sample with 50 wt.% Gly/κ-Cg and 20 wt.% NdTrif3/κ-Cg, (1.03 x 10−4, 3.03 x 10−4, and 1.69 x 10−4 S cm−1 at 30, 60, and 97 ºC, respectively).”
Comment 4
In Fig. 5, the Tg-s value all the doped samples demonstrate a non-linear variation of the ionic conductivity with temperature, a behavior typically found in disordered electrolytes. Please explain the reason in details.
Answer
We understand the comment.
Changes included
Please see answer to comment 5 of Reviewer #2.
Comment 5
What is “inner filter’’ effect? Please explain it in details.
Answer
The inner-filter effect denotes donor-acceptor radiative energy transfer, in this case the emission from the host is absorbed by the metal ion.
Changes included
The text was revised to clarify this point.
“Moreover, the observation in the excitation spectra of Figure 7(c,d) of self-absorptions in the Cg emission indicates the presence of Cg-to-Nd3+ radiative energy transfer, i.e., the Cg emission resonant with the Nd3+ intra-4f lines is absorbed by the metal ions and, subsequently, converted into f–f emission. Such radiative energy transfer has been previously observed in Nd+3 and was designated the ‘‘inner filter’’ effect [50].
Round 2
Reviewer 1 Report
The authors replies are satisfactory.
Reviewer 2 Report
The authors have adequately addressed my concerns with the paper and I now recommend it for
publication.
Reviewer 3 Report
Dear Editor:
According to revised version, It can be accepted and published in molecules journal.
Sincerely yours.